# Fluorescence-Guided Surgery in the Surgical Treatment of Gliomas: Past, Present and Future

**DOI:** 10.3390/cancers13143508

**Published:** 2021-07-13

**Authors:** Rosa Sun, Hadleigh Cuthbert, Colin Watts

**Affiliations:** 1Department of Neurosurgery, Queen Elizabeth Hospital, Birmingham B15 2GW, UK; C.Watts.2@bham.ac.uk; 2Institute of Cancer and Genomic Sciences, University of Birmingham, Birmingham B15 2SY, UK

**Keywords:** 5-aminolevulinic acid, glioma, glioblastoma, fluorescence-guided surgery, neurosurgery, survival, extent of resection

## Abstract

**Simple Summary:**

Gliomas are aggressive central nervous system tumours. The emergence and recent widespread adoption of 5-aminolevulinic acid and fluorescence guided surgery have improved the extent of resection, with implications for improved survival and progression-free survival. This review describes the history, rationale and mechanism behind the use of 5-aminolevulinic acid and fluorescence-guided surgery. We also discuss current limitations and future directions for this important adjunct to glioma surgery. This review aims to provide readers with an up-to-date overview and evidence base on this important topic.

**Abstract:**

Gliomas are central nervous systems tumours which are diffusely infiltrative and difficult to treat. The extent of surgical resection is correlated with improved outcomes, including survival and disease-free progression. Cancerous tissue can be directly visualised intra-operatively under fluorescence by administration of 5-aminolevulinic acid to the patient. The adoption of this technique has allowed surgeons worldwide to achieve greater extents of resection, with implications for improved prognosis. However, there are practical limitations to use of 5-aminolevulinic acid. New adjuncts in the field of fluorescence-guided surgery aim to improve recognition of the interface between tumour and brain with the objective of improving resection and patient outcomes.

## 1. Diffusely Infiltrative Nature of Gliomas

Gliomas are intrinsic tumours of the brain, accounting for the majority of malignant intra-axial central nervous system lesions [1]. The presence of the blood–brain barrier poses challenges for successful penetration of therapeutic agents compared to other systemic cancers. Surgical resection followed by concomitant and adjuvant chemoradiotherapy is the mainstay of treatment but sadly the vast majority of patients will re-present with disease progression and eventually succumb [2].

The 2016 revision of the WHO 2007 classification of brain tumours [3] has incorporated the most recent evidence from a rapidly expanding body of molecular data. Identification of succinct tumour populations by molecular markers, rather than through conventional morphology, has facilitated a greater understanding of how gliomas evolve and revealed additional prognostic information to guide clinical management and patient counselling.

The inherently infiltrative nature of gliomas through the surrounding brain parenchyma poses a particular challenge for neurosurgeons and neuro-oncologists. Delineation between tumour and normal brain parenchyma is often difficult to achieve both pre-operatively and intra-operatively. Despite modern advances in imaging, it is often difficult to differentiate between peri-lesional oedema and non-enhancing tumour on magnetic resonance imaging (MRI) [4], as demonstrated in Figure 1.

Glioma cells invade through the extra-cellular matrix along white matter tracts and perivascular spaces. This invasive growth occurs through a combination of changes induced in its microenvironment including modification of the stromal architecture, active degradation of normal brain matrix and immune modulation to bypass surveillance, detection and destruction [5]. Infiltration of tumourigenic cells at the tumour–brain border can be visualised directly through in vivo fluorescent imaging techniques [6]. It appears that two types of invasion occur here: a fast and non-directional spread coupled with a slow but directional infiltration characterised by leading and following tumour cell populations [6]. It is due to the need to capture these outermost invading cells that the planning of radiotherapy usually has a planning target volume that is 0.3–0.5 cm beyond abnormal FLAIR areas on imaging [7].

## 2. Extent of Resection and Correlation with Survival and Disease-Free Progression

The importance of an adequate resection, removing as much tumour tissue as is safely possible without causing a neurological deficit to the patient, has long been recognised [8]. The threshold for which neurological compromise becomes unacceptable is an important pre-operative consideration that warrants careful discussion between the operating surgeon and the patient. An increasing amount of evidence in recent years has confirmed the link between extent of resection, survival, and disease-free progression [9,10,11,12,13,14]. With the aid of new technological advancements such as intra-operative neuro-monitoring [15,16] and tractography [17,18], we are now able to stretch the limits of safe surgical resection, challenging and evolving our understanding of ‘maximal safe resection’ in the modern surgical era [19,20].

Increased resection rates also increase tumour response to chemo-radiotherapy [21]. This growing body of evidence is becoming increasingly recognised and integrated into guidelines for optimal surgical management of glioma patients [22,23,24].

The total volume of resection to achieve an optimum trade-off between safety and good clinical outcome remains debated [9]. The general consensus is that clinical improvements correlate with the amount of diseased tissue removed [8,25]. In particular, gross total resections are most likely to be associated with improved clinical outcomes in terms of both disease-free progression and overall survival [12,13,14]. However, the majority of reports describe the benefits of cytoreduction in terms of resection above a certain percentage of pre-operative tumour volume. Minimum resection margins have been reported to be over 78% [25] and over 70% [26]. An alternative perspective is that it is the volume of residual disease rather than the percentage volume reduction that is the critical factor [27]. The debate is further complicated by a lack of standardised and validated criteria to quantify resection, which typically varies from study to study.

If indeed there is a “surgical minimum” threshold that confers clinical benefit it is likely to differ between tumours with different molecular characteristics, as well as patient factors and oncological treatment. Various molecular markers [28,29] demonstrate the vast heterogeneity of low-grade gliomas, of which extent of resection is only one contributing factor towards predicting prognosis. However, extent of resection has been linked to improved clinical outcomes for all grades of gliomas, though this appears to have a stronger evidence base in higher-grade tumours [8,30,31].

## 3. Fluorescence-Guided Surgery and the Introduction of 5-ALA

Over the last two decades, there has been an exploration of intra-operative adjuncts designed to optimise safe glioma resections, led by fluorescence-guided surgery (FGS). Fluorescence refers to the emission of light by a substance which has absorbed electromagnetic radiation. It is commonly seen throughout nature and has a number of industrial uses including the ubiquitous fluorescent light bulbs that grace most hospitals. The first application of fluorescence in surgery dates back to 1948, when fluorescein was used in the localisation of intracranial tumours [32]. In this study, patients were given an intravenous injection of fluorescein, and the presence of fluorescence was used to confirm tumour presence and to examine the extent of surgical resection. This represents the first published example of the use of fluorescence-guided neurosurgery.

More recently, intra-operative fluorescence guidance using 5-aminolevulinic acid (5-ALA) has become widely trialled and adopted [30,33,34]. 5-ALA is a naturally occurring precursor to haem and is converted to protoporphyrin IX (PpIX) within cells along the haem biosynthetic pathway [35]. Administration of 5-ALA, a non-fluorescent pro-drug, leads to intracellular accumulation of PpIX [36,37]. PpIX highly absorbs light in the violet spectral range (380–420 nm) and fluoresces at the red end of the visible spectrum (620–710 nm, with a peak at 635 nm), which can be visualised intra-operatively with a microscope fitted with appropriate optical filters. Central core areas of tumour tissue fluoresce a bright ‘red’, surrounded by margins of what appears a lighter ‘pink’ fluorescence reflecting infiltrating disease [33].

The exact mechanism by which 5-ALA specifically targets tumour cells remains unclear. 5-ALA delivery into proliferating cells is facilitated by disturbance of the blood–brain barrier in tumours [33]. Intracranial tumours have a higher rate of cellular proliferation and metabolic activity compared to adjacent brain parenchyma. There are differences in mitochondrial metabolism and enzymatic activity, especially a reduction in ferrochelatase activity, which converts PpIX to haem [38]. Once PpIX is synthesised there is also a reduction in porphyrin efflux out of tumour cells [39]. All of these factors contribute to a build-up of PpIX within tumour cells compared to normal tissue.

5-ALA is administered orally at a dose of 20 mg/kg, 4–6 h prior to surgery. Tumour tissue can be directly visualised by emitting pink fluorescence under blue light, with 85% sensitivity and 100% specificity reported in the initial piloting study [33]. Similar or even higher values for the sensitivity of 5-ALA for malignant glioma tissues have been consistently demonstrated as familiarity with the technology has grown [40], with up to 95% sensitivity reported in one study [41]. Specificity for fluorescence in predicting malignant glioma tissue has a wider degree of variance, although in most studies this was found to be above 70% [42,43,44]. However, much of this data is retrospective, and there is limited prospective data. The degree of fluorescence has been shown to correlate both with the density of tumour tissue [45] and tumour aggressiveness [44], for which intra-operative visualisation of weak fluorescence can achieve a positive predictive value of up to 99% for tumour identification [42]. The degree of 5-ALA fluorescence is dependent on the amount of malignant tissue within a tumour. The histological grade of glioma will thus affect whether intra-operative fluorescence is observed. This has been shown to vary from 95.4% in glioblastomas (WHO grade IV) to 24.1–26.3% in grade I and grade II gliomas [46].

Figure 2 illustrates areas of ‘strong’ and ‘weak’ fluorescence visualised under blue light, compared to under standard operative white light. Solid red fluorescence signifies a higher concentration of malignant tumour cells, whilst a lighter shade of pink is usually present around the tumour borders, indicating infiltration of tumour into brain parenchyma [40]. The ability to directly see malignant tissue throughout the surgical process helps to compensate for brain shift that can limit the value of intra-operative imaging techniques as surgery progresses [47,48].

## 4. Impact of Using 5-ALA

The use of 5-ALA improves the extent of resection, demonstrating that real-time intra-operative visualisation of malignant tissue can lead to wider resection margins that often extend beyond the limit of contrast enhancement on pre-operative imaging [49,50]. This is explained by the ability of 5-ALA fluorescence to highlight tumour tissue outside the blood–brain barrier. The addition of fluorescence was also shown to highlight areas of tumour tissue that would have otherwise escaped attention under standard operative white light [42,51].

The now widely accepted notion that gross total resections and radiologically complete resections of enhancing tumour carry the most favourable survival and disease-free progression outcomes has transformed clinical practice in recent years. The increased adoption of routine 5-ALA use worldwide is a testament to this new method of surgical thinking.

Reports on how much additional improvement in EOR achieved with 5-ALA vary, though since 2006 there has been an increasing trend of higher gross total resections demonstrating growing familiarity with the technique. Stummer reported complete resection of 65% in patients undergoing 5-ALA aided resection, versus 36% in those operated under standard white light conditions. Figures as high as 93% for GTR > 98% and 100% for GTR > 90% have been reported [52].

The prospective trial data [30,34] has led to the approval of 5-ALA in the European Union for demonstration of achievement of higher rates of gross total resection, compared to patients who were operated on using standard white light. Encouragingly, the results also demonstrated higher 6-month progression-free survival for patients undergoing resection with 5-ALA and prolonged survival for patients achieving complete resection with the aid of 5-ALA [14,30,53]. However, this body of literature is methodologically heterogenous, and currently there are no meta-analyses of available literature which can delineate the true degree of survival advantage 5-ALA offers. To truly answer this question, multi-centre coordinated efforts gathering large-scale prospective evidence across all glioma subtypes is needed. The up-and-coming clinical trial platform Tessa Jowell BRAIN MATRIX [54] provides a ‘hub’ for centralised co-ordination, recruitment and analysis of future trial data. Platforms such as the BRAIN MATRIX are devoted to integration of clinical and biological data and could very well hold the key to answering these questions in the near future.

Nearly a decade after European approval, 5-ALA use was approved by the FDA as an intraoperative optical imaging agent; the relative delay here was due to the initial interpretation of 5-ALA as a therapeutic rather than adjunctive agent, therefore requiring a novel regulatory process [55].

## 5. Limitations of 5-ALA

There are practical limitations of intra-operative 5-ALA use. Normal appearances during 5-ALA-assisted surgery are of a core of highly fluorescent ‘red’ tumour, surrounded by a rim of more faintly fluorescing ‘pink’ tumour which reflects the ‘infiltrating zone’ of a tumour. The main factor limiting achievement of gross total resection is resection of this ill-defined zone. Furthermore, the majority of resected glioblastomas undergo local recurrence at or near to the previous resection cavity [56,57]. Potential clinical gains by improving the resection of disease in this infiltrating zone should therefore be a priority of intra-operative surgical adjuncts.

However, infiltrating tumour is not always reliably identified by 5-ALA. There are lower levels of PpIX and fluorescence in the infiltrating zones of tumours [58]. Non-fluorescing tissues at the resection borders frequently contain tumour cells [59], and the negative predictive value of non-fluorescing samples is low [60,61], indicating that a lack of visible fluorescence does not reliably indicate normal brain tissue. False positive fluorescence also occurs in areas of normal brain parenchyma, possibly due to disruptions in the blood–brain barrier or leakage of PpIX from tumour cells into the extracellular fluid [40,62].

The utility of 5ALA as a surgical adjunct is also much more limited in low-grade gliomas (LGGs). With modern treatment regimes, patients with LGGs may survive for many years. If they are exposed to morbidity from surgery, this can lead to long-term disability and reduced quality of life [63,64,65]; the need for safe resection with minimisation of resulting neurological deficit is therefore of paramount importance.

Limited visible fluorescence in LGG is well recognised [33], with recent data suggesting the sensitivity of fluorescence in LGG is just 16% [66]. Smaller studies even report 0% macroscopically visible fluorescence [67]. Where 5-ALA-induced fluorescence is observed, it may be a marker for malignant transformation [68]. The reasons for this dramatic difference in uptake between high- and low-grade gliomas remain unclear; it is proposed that they are due to differences in cellular metabolic activity, membrane transport and local blood–brain barrier permeability [69].

Finally, 5-ALA has financial implications which may hinder adoption in some countries. The cost of administering 5-ALA along with the specialist fluorescence intra-operative equipment required has been quoted to approximate £1 million per year for a neurosurgical unit [23]. Cost-effectiveness analysis has associated 5-ALA use to the equivalent of €9021 per quality-adjusted life year in Spain [70] and $12,817 in the USA [71].

## 6. Refining the Use of 5-ALA

New adjuncts in the field of fluorescence-guided surgery aim to improve recognition of the interface between tumour and brain with the objective of improving resection, minimising damage to surrounding normal brain, and improving patient outcomes.

The practical application of 5-ALA has changed little since its introduction and approval by the EMA in 2007, and the intra-operative evaluation of fluorescence still relies on subjective evaluation with the naked eye. The degree of visualised fluorescence is related to the degree of PpIX accumulation and therefore the density of tumour cells [72]. However, it can also be impacted upon by numerous other factors, including: (1) optical properties of the tissue, such as absorption and scattering, (2) technical properties of the microscope, including viewing distance and angle and (3) photobleaching and photoproducts [73]. Photobleaching refers to the process by which PpIX fluorescence deteriorates with light exposure. This process also leads to the formation of numerous photoproducts which display significant spectral overlap with PpIX [74], which can confound measurements of PpIX concentration.

There is scope to refine and enhance the current use case of 5-ALA, either by improving differentiation between areas of minimal fluorescence and normal brain or by enhancing the degree of fluorescence, for example with laser excitation [75]. Current clinical research efforts focus predominantly on improving the identification and quantification of fluorescence. A quantitative measurement of fluorescence would allow for correction of the distorting effects of optical tissue properties and spectral unmixing of PpIX fluorescence from the fluorescence of its photoproducts and surrounding tissues. The implication is that better contrast detection between visible and non-visible fluorescence may allow more accurate delineation of the tumour margins.

Numerous groups [60,76,77,78,79] have applied spectrometric methods to assess weakly-fluorescing tissue at tumour margins, most commonly using handheld intra-operative spectroscopic probes. These techniques can identify significant levels of PpIX accumulation that are not visible to the naked eye. This gain in sensitivity may translate to more accurate tumour identification and improved extent of resection [60]. Richter et al. have integrated the use of spectrometric probes into their tumour resection routines and shown that PpIX fluorescence was seen in 67% of areas where there was no visible fluorescence under the microscope [79]. These benefits may also extend to resection of low-grade gliomas in which 5-ALA fluorescence is currently of limited value. Valdés demonstrated that a sizable fraction of LGGs which did not visibly fluoresce still accumulated levels of PpIX significantly above normal brain tissue raising the possibility that 5-ALA may retain diagnostic use in LGGs with more sensitive methods of detection [80].

Spectroscopic techniques are providing promising results in terms of improving sensitivity of PpIX fluorescence. However, there are limitations to this approach. There is currently no standardised ‘threshold level’ for what differentiates tumour from normal tissue. While sensitivity for PpIX fluorescence does improve, this has yet to translate into compelling evidence that these techniques improve rates of gross total resection and patient outcomes. There are also operative considerations; the use of handheld probes that only sample a small area of tissue may be disruptive to the surgical workflow, and its applicability to an entire resection cavity is limited. Widefield techniques that can interrogate the entire surgical field are limited in their ability to provide instantaneous quantification that would allow the surgeon to respond in real time [73].

## 7. The Future of Fluorescence-Guided Surgery

Fluorescence-guided surgery is a rapidly developing field; a recent review identified 39 new agents undergoing 85 clinical trials for all tumours [81]. The ideal fluorescent agent has a number of desirable properties: high selectivity for tumour cells, high tumour/background ratio (i.e., high specificity) to improve contrast between tumour and normal tissue, low toxicity and few contra-indications and simple use and delivery.

While macroscopically similar, glial tumours and normal brain parenchyma differ significantly in terms of their microscopic structure and behaviour. These differences can be exploited to provide contrast between the tissue types, and this is reflected in the diverse range of techniques and targets being used by novel fluorescent probes. There are a number of novel fluorescent probes undergoing early clinical trials for use in glial tumours (see Table 1) and a host of preclinical trials exploring other agents [82]. These aim to improve selectivity for tumour cells by targeting tumour-specific ligands or exploiting tumour-specific enzyme expression. Many of these novel probes have emission spectra in the near-infrared range (NIR); this is advantageous as autofluorescence and optical distortion are at a minimum in this wavelength [83].

EGFR is overexpressed in a majority of primary glioblastomas [96] but not in the normal brain parenchyma, making it an attractive target for a number of fluorescent antibody-based probes [97,98,99,100]. Both ABY-029 and panitumumab-IRDye800CW have demonstrated greater tumour/background contrast ratios and improved spatial definitions of tumour margins than 5-ALA [97,98].

Alternatively, improved contrast between tumour and normal cells can be achieved via use of an activatable probe. This would eliminate the background signal of normal tissue. 5-ALA and PpIX preferentially accumulate in tumour cells; however, the synthesis of PpIX happens in all cell types as it is part of normal biosynthetic pathways. Activatable fluorescent probes remain quenched until they are activated in specific ways within tumour cells. LUM015 is a protease-activated fluorescent probe undergoing investigation in a number of cancers, including glioma. It consists of a fluorescence-quenching molecule and a fluorophore bound by a peptide backbone [101]. This peptide is cleaved by cathepsins, a group of protease enzymes highly upregulated in cancer cells. By exploiting tumour-specific modes of activation the hope is these agents will reduce background signal and therefore improve the tumour/background ratio.

These findings also open the door to intriguing applications of fluorescent agents. EGFR-targeting fluorescent probes have demonstrated improved accuracy over 5-ALA in EGFR-positive tumours but not in wildtype tumours [97]. This raises the possibility of tailored use of surgical adjuncts to specifically target known tumour biomarkers in each individual patient. Alternatively, combined use of an EGFR-targeting fluorophore and 5-ALA together provided better delineation of tumour margins in orthotopic models of glioblastoma, raising the possibility of combined use of complimentary agents in FGS [97]. The use of combined fluorescent agents has been trialled with 5-ALA and Photofrin (a photosensitiser that, like PpIX, fluoresces in the red spectrum after exposure to blue-violet light) [102]. This study demonstrated enhanced rates of resection and survival; however, it also employed a spectrometric probe to assess resection margins, and the study group also received photodynamic therapy; hence, the isolated impact of combined fluorescence agents alone on extent of resection cannot be commented on.

An important consideration with the use of multiple fluorescent agents is one of safety. The previous study did not raise any specific safety concerns with combination therapy; however, liver function tests (which can become deranged after 5-ALA administration) were not routinely monitored, and the power of the study would be insufficient to detect rare adverse events. Further, it is a reasonable assumption that the use of dual fluorescent agents may also increase the rate of false positives in fluorescing tissues, potentially increasing the risk of inadvertent resection of normal brain and its associated complications. Studies designed specifically to assess the safety of combined therapy will therefore be required.

## 8. Summary

The evidence base for routine use of 5-ALA in glioma surgery is increasing; however, the quality of clinical data remains low [66]. To date, there has only been one randomised controlled trial on the use of 5-ALA, Stummer’s original 2006 study in the evaluation of its role in the extent of resection [30]. Data detailing the benefits of 5-ALA in improving the extent of resection as well as overall survival are further limited by selection bias in a vastly heterogenous literature base. As a consequence, the guidance on when to use 5-ALA remains loosely defined. NICE guidance in the UK (www.nice.org.uk/guidance/NG99, accessed on 1 May 2021) supports its use in patients with suspected high-grade gliomas for which complete resection is possible [23]. Similarly, regulatory discussions by FDA [55] license the use for 5-ALA in glioma surgery based on its potential for EOR maximisation, although it does not provide guidance on when 5-ALA should be used.

Fluorescence-directed resection comprises part of an armoury of operating tools used in neuro-surgical oncology. It improves the contrast between tumour and normal tissue, thereby improving rates of resection. However, it should not be used in isolation because the overriding surgical objective is to maximise resection whilst minimising the risk of any neurological deficit to the patient. To this end, fluorescence-directed resection must be combined with other intra-operative adjuncts, rigorous pre-operative planning and a detailed understanding of functional neuroanatomy.

## 9. Conclusions

The addition of 5-ALA and FGS has proved an invaluable tool in helping surgeons worldwide in achieving maximal safe resections for glioma surgery. The ability to directly visualise malignant tumour tissue intra-operatively has strong implications for maximising extent of resection, improving survival and progression-free survival. Despite limitations by the inherent heterogeneity of gliomas and patient populations, the body of literature behind its influential role in glioma surgery is growing. Recent advancements in refining the use of 5-ALA as well as discoveries in novel fluorescence techniques hold future promises for FGS. It is our hope, as well as that of the wider scientific community, that this important field will continue to grow, flourish and benefit patients in generations to come.

## Figures and Tables

**Figure 1 cancers-13-03508-f001:**
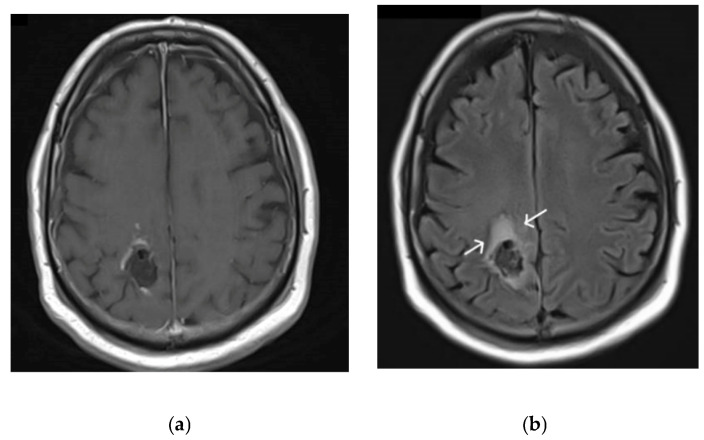
(**a**) T1 gadolinium enhanced MRI sequence demonstrating tumour. (**b**) T2 FLAIR MRI sequence demonstrating significant perilesional oedema, illustrated by the white arrow, where infiltrative tumour is present.

**Figure 2 cancers-13-03508-f002:**
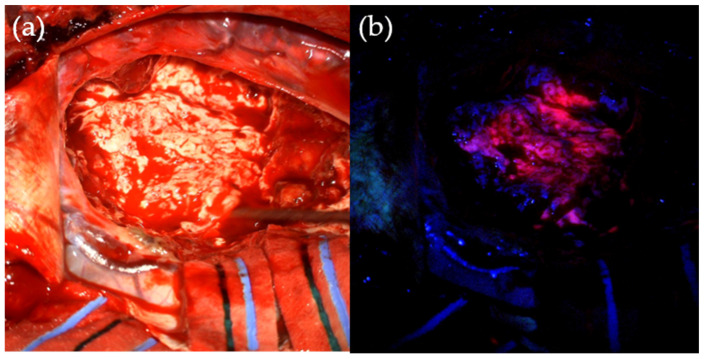
Intra-operative use of 5-ALA. (**a**) Tumour under standard operating white-light. (**b**) Pink fluorescence of tumour tissue, shown after 5-aminolevulinic acid administration and under operating microscope blue light filter.

**Table 1 cancers-13-03508-t001:** Summary of novel fluorescent agents currently undergoing clinical trials in glial tumours.

Agent	Mechanism	Fluorophore	Peak Emission Wavelength	Clinical Trial(s)
LUM015	Protease-activated fluorescent probe	Cy5	675 nm [84]	Phase 1—NCT03717142 [85]
Demeclocycline	Fluorescent antibiotic which accumulates in tumour cells	Demeclocycline	535 nm [86]	Phase 1—NCT02740933 [87]
ABY-029	EGFR-targeting fluorophore-labelled ‘affibody’	IRDye800CW	789 nm [88]	Phase 0—NCT02901925 [89]
Panitumumab-IRDye800CW	EGFR-targeting fluorophore-labelled antibody	IRDye800CW	789 nm [88]	Phase 2—NCT03510208 [90] Phase 2—NCT04085887 [91]
Cetuximab-IRDye800CW	EGFR-targeting fluorophore-labelled antibody	IRDye800CW	789 nm [88]	Phase 2—NCT02855086 [92]
BLZ-100	Fluorophore-labelled chlorotoxin	ICG	820 nm [88]	Phase 1—NCT02234297 [93]
BBN-IRDye800CW	Fluorophore-labelled peptide, targets gastrin-releasing peptide receptor	IRDye800CW	789 nm [88]	Phase 1 in HGG—NCT02910804 [94] Phase 1 in LGG—NCT03407781 [95]

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
