# Peer review of "Fluorescence-Guided Surgery in the Surgical Treatment of Gliomas: Past, Present and Future"

_cancers, 2021, doi:10.3390/cancers13143508_

Round 1

Reviewer 1 Report

The article is a short review on the topic of fluorescence photosensitizers that can be used during glioma resection, discussing the problems with brain tumor treatment and resection, pros and cons of the current 5-ALA usage and extending the discussion to the photosensitizers that are under investigation. The paper is well-written and easy to follow. It covers the major aspects of 5-ALA application as the common method on the topic in neurosurgery with a fair, although not complete, coverage of the literature and only minor errors on the facts known on this topic. Reviews on the 5-ALA fluorescence guidance are already covered well in the literature, therefore this paper is not novel in that aspect, however, it does cover some interesting perspectives on the topic and provides an update on the potential new photosensitizers that can increase the efficacy of the fluorescence guided brain tumor resection in future.

Detailed comments:

Line 42: Reference 6 does not expand on the imaging technology, in case the purpose of this sentence is to mention the in-vivo microscopy technology. 

Lines 85-94, Fluorescein has been used in a wider range than only one study mentioned. Moreover, Photofrin has also been used in neurosurgery which can be mentioned.

Lines 99-100: “PpIX fluoresces in the violet spectral range (380-420nm) and gives off visible light at the red end of the visible spectrum (620- 640nm),..”

Correction: PpIX highly absorbs light in the violet spectral range (380-420)- The absorption spectrum is wider than this, however, this UV-V range is commonly used for excitation.

Fluoresce= gives off light, PpIX fluroescence emission is in 620-710nm, however the strongest red color is due to the peak at 635nm.

Line 103: Although ‘pink’ is understandable, there is in principle no pink fluorescence. The PpIX fluorescence is red, which when mixed with the blue light reflection (and some yellow due to autofluorescence), looks pink to the eye if the dominance of the red color is low.

Line 186: FPs cannot be due to tissue autofluorescence, since that has a known spectral range in the yellow spectral region with a very low intensity in the red region which is well distinguished.

Line 194-200: The authors mention a low uptake of ALA in the low-grade tumors as a limitation of 5-ALA. However, there is no motivation why any other substance should not have this limitation. Please also see the below paper regarding the sensitivity issues of the microscope where more can be detected with technologies that have a higher sensitivity:

 https://thejns.org/view/journals/j-neurosurg/123/3/article-p771.xml

Lines 223-224: Please clarify on “However, there is no standardization of this approach, and its applicability to an entire resection cavity is limited.”

Currently, the absolute measurements have not been possible to compare among different publications, however there have been concrete efforts to quantify the PpIX concentration using spectroscopic methods, addressing technical measurement standardization, the main work is the method proposed by Kim et al. which has been widely used in the papers who used quantified methods so far. https://doi.org/10.1117/1.3523616

Standardization will anyways need to be addressed with any other photosensitizer and thus is not only a limitation or refining point needed for 5-ALA. Please also see the below paper if useful for your purpose regarding standardization of clinical studies: https://www.ncbi.nlm.nih.gov/pmc/articles/PMC6739423/

Line 223-226: There are also publications on wide field quantitative imaging of PpIX that can be relevant as alternative technologies.

https://www.frontiersin.org/articles/10.3389/fsurg.2019.00031/full

Please also see the below papers where a workflow is suggested for use of the point-wise approaches implemented in the tumor cavity.

https://www.ncbi.nlm.nih.gov/pmc/articles/PMC6616084/

https://www.sciencedirect.com/science/article/pii/S1572100016301429

Section 7:

Table 1: Please give more details in the heading of the third column.   e.g. Clinical trial/Literature

PpIX fluorescence in the visible optical range is very suitable for direct visualization during operation. Could the authors comment of the fluorescence of the substances in the table by adding the emission wavelength range?

Lines 242-247: This part can be expanded on, both on the initial clinical implications/results and on the comparison with 5-ALA. 

The authors often have used ‘tumor/background ratio’. The term is not entirely correct: 1- it should be tumor/normal ratio or tumor/gliotic tissue or tumor/edema if background refers to area around the tumor  2- Is there any reason why the authors have not used ‘specificity’ instead?

Lines 259-262. A combination of the markers often leads to a higher accuracy, however, administration of the two drugs simultaneously might not be clinically desirable. Please comment on the safety aspects of such an approach. 

Reviewer 2 Report

In the review, Sun et al have discussed the role of 5-ALA for glioma surgery. The authors have summarized the application FGS in differentiating the tumors from normal tissues. The overall theme of the review is very interesting and is of interest to the readers of Cancers. The reviewer suggests minor changes to improve the article further.

Major Comments.

  1. The title of article mentions about FGS from perspective of Gliomas. However, the article is mostly discussing 5-ALA. Other fluorophores like ICG (doi.org/10.3171/2020.10.FOCUS20782) have also been used in clinical trials with gliomas. It would be worthwhile to mention those studies.
  2. The authors have used the term glioma generally, whereas gliomas could be further divided in multiple different categories. The amount of fluorescent signal generated from 5-ala and other fluorophores are generally related to it (10.14791/btrt.2019.7.e38).
  3. In the limitations section, it is advised to discuss the wavelengths of light and absorption/scattering in tissues.
  4. The authors should also discuss the instrumentation and the lasers associated with FGS.
  5. In gliomas, Blood brain barrier is one of the main challenge that has reduced impact of external agents, specially large molecules like antibodies.

Minor Comments.
1. Not necessary but it would increase the scope of the review the authors discuss other treatment option for gliomas and the success rate and compare it with FGS.

2. The authors might want to add another section with pre-clinical probes that are being used to treat gliomas.
